# Optimisation of Physical and Chemical Treatments to Control Browning Development and Enzymatic Activity on Fresh-cut Apple Slices

**DOI:** 10.3390/foods9010076

**Published:** 2020-01-09

**Authors:** Luna Shrestha, Boris Kulig, Roberto Moscetti, Riccardo Massantini, Elke Pawelzik, Oliver Hensel, Barbara Sturm

**Affiliations:** 1Department of Agricultural and Biosystems Engineering, University of Kassel, 37213 Witzenhausen, Germany; 2Department for Innovation in Biological, Agro-food and Forest systems DIBAF—University of Tuscia, 01100 Viterbo, Italy; 3Quality of Plant Products Section, Department of Crop Sciences, Faculty of Agriculture, Goettingen University, D-37075 Goettingen, Germany; epawelz@gwdg.de

**Keywords:** fresh-cut apple, anti-browning treatment, peroxidase, polyphenol oxidase, optimisation, browning

## Abstract

Optimisation of processing time and pre-treatments are crucial factors prior to apple drying to produce a high-quality product. The purpose of the present study was to test the utility of physical (hot-water, HWB and steam blanching, SB) and chemical (1% ascorbic acid, AA; and 1% citric acid, CA) treatments, alone or in combination in reducing surface discolouration as well as oxidative enzyme activity in apple slices (cv. Golden Delicious and Elstar) exposed to air at room temperature for 0, 30 and 60 min. The total colour change (ΔE) for Golden Delicious was equal to 2.38, 2.68, and 4.05 after 0, 30 and 60 min of air exposure, respectively. Dipping in AA solution (1% *w*/*v*) was found to be the best treatment to limit surface discolouration of both apple cultivars. The best heat treatments to inhibit polyphenol oxidase/peroxidase enzymes activity were 70 °C HWB for Golden Delicious and 60 °C HWB for Elstar slices, both in combination with a solution of 1% AA and 1% CA. The tested apple cultivars were found to require different treatments at minimum ambient air exposure to obtain the best surface colour condition.

## 1. Introduction

Processed product consumption trends are rapidly increasing in different food practices due to nutritional, functional and convenience aspects [1]. For example, frozen or dried apple slices are typically used in bakery or cereal applications to meet nutritional and food security needs [2,3]. Therefore, the apple processing industry aims to design products characterized by sensory and nutritional properties similar to those of fresh produce to fulfil consumer demands. The fresh-like products are made by applying different chemical additives during processing operations. Table 1 provides a list of pre-treatments that have been previously applied in fresh-cut apple to minimize the discolouration and enzyme activities (EA). Sulfur dioxide (SO_2_) or sulfites (SO_3_
^2−^) are added to foods to control enzymatic and non-enzymatic browning, microbial growth, as reducing agents or antioxidants. However, they are not permitted in the organic food production sectors. This is due to adverse side effects on asthmatics and restriction by the US Food and Drug Administration [4]. 

More than 50 per cent of foods susceptible to browning are wasted due to discolorations caused by enzymatic reactions [5]. Undesirable endogenous enzymatic reactions occur during intermediate processing operations such as fruit washing, cleaning, sorting, peeling, coring and cutting [6]. Polyphenol oxidase (PPO) and peroxidase (POD) are the two enzymes responsible for the oxidative browning reactions. These enzymes are present in almost all plants, but most abundantly active in fruits such as apples [7,8]. During cutting operations, the fruit cell ruptures allowing oxygen in the air to react with the PPO and substrate thereby causing browning [9]. PPO catalyzes the oxidation of monohydroxy phenols (phenol, tyrosine, *p*-cresol) to *o*-dihydroxyphenols (catechol, dopamine, and adrenaline) and dehydrogenation of *o*-dihydroxyphenols to o-quinones producing melanins which manifest as brown colour in the foods [10,11]. Additionally, enzyme activity leads to the deterioration of nutritional quality, flavor and shelf life of the products [12]. In this context, apples are more vulnerable to browning reactions because of the presence of various phenolic compounds such as chlorogenic acid, which are oxidized to quinones by the PPO. Subsequently, quinones are polymerized with other quinones and amines to form brown pigments [13,14,15]. Phenolic compounds present in apples vary with cultivars, environmental parameters, agronomic practices, harvesting and storage conditions [11,16]. Therefore, it is essential to control the pre-processing conditions accurately to avoid the discolouration caused by enzymatic activities, losses of nutrients, weight and colour change [17].

Consumer awareness about the different types of ingredients applied in processed foods is increasing [30]. At the same time, the idea of organically produced food is growing in most part of the world because people are getting more concerned about their health and environment aspects [31]. Therefore, the production of diversified food products without using chemical ingredients such as sodium metabisulphate has become a primary matter of concern in the food processing industries. Moreover, new advanced techniques such as high hydrostatic pressure (HHP) have emerged to preserve the nutritional content and colour of final dried products [32,33]. However, the application of these sophisticated techniques might be challenging to small and medium (S&M) scale processing units due to lack of technical knowledge and economic cost. Therefore, application of blanching and use of ascorbic and citric acid, which are cost effective and easy to use might be helpful to minimise the browning of dried apple slices and to maintain their nutritional content. 

Previous studies have focused on several anti-browning pre-treatments such as calcium ascorbate with ultrasound to reduce the enzymatic activity in apple slices. These pre-treatments might be difficult to apply for S&M scale processing units. Thus, there is a need of technique that could be easily application and cost effective. Based on the author’s knowledge, no studies have been documented related to the combined effect of acids (ascorbic or citric acid) and blanching pre-treatment (hot-water or steam-blanching) on Golden Delicious and Elstar apple cultivars. Additionally, less attention has been given to the duration of air exposure after the pre-processing stage and most of the studies have focused on the storage duration [34,35]. This study aims to test the potential of physical and chemical treatments, alone and in combination, and to find out the best pre-treatments to reduce the browning development of apple slices from two cultivars exposed to air at room temperature. 

## 2. Materials and Methods 

### 2.1. Raw Material

Apples of cv. Golden Delicious and Elstar originating from Germany were purchased from a local Edeka supermarket (Goettingen, Germany) and a farm of the University of Kassel, Hessische Staatsdomäne Frankenhausen, (Grebenstein, Germany), respectively. The apples were stored at 4 °C until use in experiments.

### 2.2. Reagents and Chemicals

All chemicals and reagents used in the experiments were of analytical grade. Disodium phosphate (Na_2_HPO_4_), monosodium phosphate (NaH_2_PO_4_), dipotassium phosphate (K_2_HPO_4_), monopotassium phosphate (KH_2_PO_4_), catechol, guaiacol, hydrogen peroxide, polyvinyl-polypyrrolidone (PVPP) and Triton X_100_ were purchased from Carl Roth (Carl Roth GMBH + Co.KG, Karlsruhe, Germany). Deionized water was used in the preparation of solutions and buffers.

### 2.3. Sample Preparation

Apples of uniform size and colour were selected for the experiments. The apples were cored using a stainless-steel corer of 250 mm (Lurch AG, Hildesheim, Germany) and sliced to 5 mm thickness using an electrical slicer (Graef, Allesschneider Vivo V 20, Arnsberg, Germany). Apple slices were cut into equal outer diameter using a cookies dicer of 620 mm diameter (Lurch AG, Hildesheim, Germany). The sample preparation was performed at ambient temperature for 2 min.

A total of 192 apples were used for the experiments. Each apple was sliced to 3 slices and each slice was exposed to air for 0, 30 and 60 min respectively. The average values of measurements on 3 apple slices were calculated for each run. Colour determination and PPO/POD extraction were carried out in triplicate for the three exposure times.

### 2.4. Sample Characterization

A digital refractometer (OPTECH GmbH, München, Germany) was used to measure total soluble solid content (TSS) in the two apple varieties. TSS was expressed as a percentage. The pH of the juice extracted from the apples was measured using a digital pH meter (Inolab digital pH meter, Wissenschaftlich-Technische Werkstätten, Weilheim, Germany). Titratable acidity (TA) was determined using a standard titration method described by Bartolini, Viti and Ducci, [36] TA was expressed as gram (g) of malic acid per 100g fresh weight (FW). Moreover, the TSS/TA ratio was also calculated by dividing the TSS values by TA values. 

### 2.5. Anti-Browning Treatments

Slices of both cultivars were treated using blanching and dipping treatments, alone and in combination. The blanching treatment consisted of (1) hot-water blanching (HWB) at 50, 60 and 70 °C, (2) steam blanching (SB) at 65, 75 and 85 °C, and (3) control (no blanching). Both blanching treatments were performed in a water bath with a maximum capacity of 22 L with interior dimension of 350 × 220 × 290 mm (Memmert GmbH Co, WNB22, Schwabach, Germany). Steam blanching involved injecting steam through an equally stainless-steel perforated wire tray with stands in which apple slices were placed. During both water and steam blanching treatments temperature was monitored with a K292 data logger thermometer (Voltcraft, Hirschau, Germany). Dipping treatments were performed using anti-browning agents by using (1) 1% AA (ascorbic acid), (2) 1% CA (citric acid) and (3) 1% AA + 1% CA solutions as well as (4) control (no dipping). All blanching and dipping treatments were carried out for 3 min [18,37]. All treated samples were exposed to air at room temperature and analytical measurements were performed at 0, 30 and 60 min of air exposure.

### 2.6. Colour Assessment

The colour of apple slices was measured with a Chroma Meter CR-400 colorimeter (Minolta, Osaka, Japan). The mode used for acquiring data was CIE Standard Illuminate C and 2° observer angle. The colorimeter was calibrated using a standard white reflector plate with values of Y (92.4), × (0.316) and y (0.3322). Four replications were carried out for each sample by performing two colour measurements on each side of the apple slice. The results were expressed in the CIELab colour space in terms of luminance (*L**), redness (*a**), yellowness (*b**). Moreover, the browning index (*BI*, Equation (1)) and the CIE 1976 colour difference (*ΔE*, Equation (3)) [38,39] were both calculated by comparing colour coordinates taken before and after treatment at 0, 30 and 60 min of exposure to air. *BI* and *ΔE** were acquired to obtain complementary information related to changes in colour during air exposure at room temperature. The *BI* indicates brown colour purity of the food products rich in sugar [40]:(1)$$BI\;=\frac{\left(x\;-\;0.31\right)}{0.172}\times \;100$$
where, *x* is the chromaticity coordinate calculated from the XYZ tristimulus values according to the following formula:(2)$$x\;=\;\frac{\left(a^{*}+1.75\;L^{*}\right)}{5.645\;L^{*}+a^{*}-3.012{\;b}^{*}}$$

The change in colour *ΔE* is calculated by using following formula:(3)$$\;\Delta E=\;{\left(\;\Delta L^{2}+\;\Delta a^{2}\;+\;\Delta b^{2}\right)}^{\frac{1}{2}}$$
where:$$\Delta L^{*}=\;L^{*}-L_{0}^{*};\;\Delta a^{*}=\;a^{*}-a_{0}^{*};\;\Delta b^{*}=\;b^{*}-b_{0}^{*}$$

The subscript ‘0’ refers to the initial colour parameters of each sample at the beginning of experiment.

### 2.7. Enzyme Activity Assessment 

PPO and POD enzyme activities were both measured with a mod. HP 8453 UV-Vis spectrophotometer system (Agilent Technologies GmbH, Waldbronn, Germany) following the protocols described by Furumo and Furutani, [41] and Zhang and Shao [42], with some modifications.

Approximately 10 g of sample were homogenized with 20 mL sodium phosphate buffer (PBS, 0.1 mol/L, pH 6.5) containing 1.5% (*w*/*v*) polyvinylpolypyrrolidone and 1% (*v*/*v*) Triton X_100_. The homogenization was performed under ice-cooling by using an Ultra Turrax T25 (IKA Instruments Ltd., Staufen, Germany). The mixture was centrifuged at 12,400× *g* for 20 min at 4 °C using a centrifuge mod. Eppendorf 5416 (Eppendorf GmbH, Hamburg, Germany). The supernatant was then filtered through a Macherey-Nagel (MN) 640 w filter paper. Finally, the crude enzymatic extract was stored at −80 °C until use. PPO enzyme activity was measured using a reaction mixture containing 1.5 mL of catechol (40 mmol L^−1^), 2.3 mL of PBS (0.1 mol L^−1^, pH 6.5) and 0.2 mL of crude enzyme.

POD enzyme activity was assayed using a reaction mixture containing 0.15 mL of guaiacol, 0.05 mL of H_2_O_2_ (1% *v*/*v*), 2.3 mL of PBS (0.1 mol l^−1^, pH 6.8) and 0.5 mL of the enzyme extract.

PPO and POD enzyme activities were measured as changes in absorbance at 420 nm and 470 nm, respectively. Absorbance values were recorded every 2 seconds for 2 min upon oxidation of the substrate catalysed by the enzyme. One unit of enzyme activity (1 UEA) was defined as an increase in absorbance of 0.001 min^−1^. 

### 2.8. Design of Experiments

In this study, a multilevel 4 factor-fractional design was chosen in a split-plot design as shown in Table 2. The study was accomplished in 64 runs in 14 main plots to describe the colour changes and enzymatic activity in apple slices from two cultivar. The software Design-Expert 10.0.5 [43] was used to formulate the design and for data analysis.

### 2.9. Statistical Analysis

Split-plot design was designated to conduct the experiments, where the Restricted Maximum Likelihood (REML) approach was adopted to estimate unbiased variance components. The split-plot design divides the experimental runs into two strata: the whole plots and subplot runs. A whole plot is a group of runs where the hard-to-change factor is set up and the individual subplot runs consists of easy-to change factors [44]. The final model (i.e., general linear model) was obtained by applying a backward selection strategy. At the beginning, a full model with all available terms up to 2nd order interactions and quadratic terms was built. The goodness-of-fit of the polynomial equation was characterized by the coefficient of determination (R^2^). The difference between the R² and adj. R² was minimized via iterative model selection, to get a robust model for prediction. All non-significant (α > 0.05) terms were eliminated stepwise from the model in compliance with the hierarchy basis. Thus, the final model contains only significant terms considering the hierarchy of the model. Graphical tools were used to identify deviation against basic model assumptions. In case of heterogeneity and non-normality of the residuals, a statistically useful and technically relevant transformation to the response variables was carried out. Furthermore, ANOVA was performed to evaluate the significant effect of each factor at a probability at 95% confidence level [45]. The coefficient estimates were determined in coded units; therefore, the actual factor levels were standardized on a range from −1 for the lowest actual level to +1 for the highest actual level for each factor. Since all estimators were computed in standardized form, the leverage of a factor was directly interpreted in the same units as the response variable. Derringer’s desirability function was applied to find the experimental conditions (factor levels) to reach and the optimal value for all the evaluated variables during the optimization procedure. Individual desirability indices for each response were determined using the fitted models and establishing the optimisation criteria. A combined desirability index was then built for all responses to balance out conflicting goals to find an overall optimum value. Desirability take values between 0 and 1, with 0 for undesirable values, and 1 representing a completely desirable value [46]. The value of desirability, however, depends on the severity of the goals being defined. It can thus be interpreted in relation and serves only as a mathematical criterion for the optimisation algorithm. In any case, the highest value reached is also the operation point at which the global goal is best reached.

Only treatments which showed residual PPO/POD enzyme activity higher than 0 were included in the experimental design. The output focused on the optimum findings and best solution regulating the settings of the targeted goal (i.e., minimising the colour change and the residual EA on apple slices). Data from sample characterization (i.e., TSS, TA, pH, and TSS / TA ratio) were analysed through the pairwise comparison t-test by using the SPSS 24.0 software [47] ( SPSS Inc, Chicago, IL, USA).

## 3. Results

### 3.1. Apple Fruit Characterization

Table 3 indicates a significant difference (*p* < 0.05) in TSS, pH, TA and TSS / TA ratio between the two cultivars. The highest TSS/TA ratio was observed in Golden Delicious cultivar. This result was consistent with findings by Jaros et al., [48] and Petkovska et al., [49], which showed Elstar cultivar to have a higher titratable acidity, lower pH and lower TSS/TA ratio. Moreover, the findings showed apple cultivars to constitute different chemical compositions consistent with the findings of Pires et al., [50], Wolfe et al., [51] and Drogoudi et al. [52]. The higher TSS/TA ratio of Golden Delicious reflects both a sweet flavour and higher susceptibility of the fruit to browning reactions during processing operations due to a higher sugar content. In addition, the chemical properties of apples (i.e., sugars) could accelerate the Maillard reactions (MR) which also contributes to browning [53]. 

### 3.2. Numerical Optimization for Colour Changes of Apple Slice Surface

In the present study, the optimal output was achieved through the simultaneous optimisation of multiple responses using the desirability function. Specifically, the goals were combined into an overall desirability index. Additionally, an individual response was shown at the optimum point. Under the optimized conditions, total desirability indices of 0.883 (Elstar) Figure 1a and 0.871 (Golden Delicious) Figure 1b were obtained.

Samples dipped in a 1% AA + 1% CA solution revealed an optimum result for slices from both cultivars minimising the brown colour development. The total colour difference (*ΔE*) was more pronounced in Golden Delicious in comparison to Elstar, after 60 min of air exposure (Figure 1b). Thus, this gives a clear indication that the two cultivars have differential susceptibility to browning caused by EA [54,55]. Moreover, results highlighted that browning development intensified when an apple slice was exposed to air for a long time.

In this study, blanching treatments were not effective in reducing the browning reactions of the investigated slices. Our findings are consistent with results from Lavelli and Caronni [22], which showed blanched apples were darker in colour than un-blanched ones. This might be due to a loss of cellular integrity due to the pre-heating treatment which enhanced the browning rate. Nevertheless, enzyme activity was inhibited by the blanching treatment resulting in a lower enzyme activity in blanched apples. the applied blanching temperatures were effective in deactivating the enzymes but degradation of several natural pigments such as flavonoids [56] might also have occurred when exposed to heat leading to browning.

Appendix A (see the Appendix A) presents the ANOVA, where the cultivars show a significant effect on PPO and POD and colour parameter (*L** and *b**) at 0 min, 30 min and 60 min air exposure timing. However, the *a** parameter at 0, 30 and 60 min was not significant for both cultivars. Water blanching showed a significant effect in 30 min and 60 min waiting time for PPO and POD activity and all CIELab colour parameters. The effect of ascorbic acid was significant on most of the responses except for *L** and *b** at 0 min air exposure timing. Citric acid and interaction effect (ascorbic acid-variety, water source-variety) were non-significant for most of the responses, except POD, thus these factors were eliminated from the model in compliance with the hierarchy approach.

### 3.3. Effect of Treatments on the Development of the CIE 1976 Colour Difference (ΔE)

As already stated, development of both *ΔE* and BI were more pronounced for Golden Delicious samples. This might be due to the higher concentration of chlorogenic acid in the Golden Delicious variety [57]. Chlorogenic acids act as a prominent substrate for PPO and POD EA, thus, accelerating the browning reaction [14,54]. Perhaps, the higher level of sorbitol in the Elstar apple partially inhibited the oxidative enzyme activity and helped to prevent browning processes [58,59].

There was no difference in colour change *ΔE* between control sample at 0 min and AA dipped sample at 0 min. However, with increase in air exposure (i.e., at 30 min and 60 min), the colour change was more pronounced in the control sample compared to sample treated with AA solution (Figure 2). The anti-browning effect of AA treatment was much more distinguishable in 70 °C HWB than for the other treatments used. Remarkably, the *ΔE* increased after 60 min of air exposure in the control samples (Figure 2c). The effect of CA did not show a significant difference in the applied treatments (Appendix A). The *ΔE* was very high when a heat treatment was applied. This can probably be ascribed to the occurrence of MR, in which the high content of sugar (mainly fructose in apple fruit) reacts in presence of heat source resulting to discolouration [58,60]. At the higher water blanching temperature, the *ΔE* increased. This might be due to a fast AA degradation when cells were disrupted, thus, MR was accelerated [61]. However, the addition of AA reduced the *ΔE* at both 50 and 60 °C HWB treatments. On the contrary, a pronounced increase in the *ΔE* in the slices (cv. Golden Delicious) was distinctive for the 70 °C HWB in combination with a dipping solution of 1 % AA.

### 3.4. Effect of Treatments on Changes in L* and a* Values

Figure 3 showed trends of luminance (*L**) for both Golden Delicious and Elstar slices before applying treatments (Figure 3a) and after applying treatments measured at 0, 30 and 60 min (Figure 3b–d). 

Figure 4a,b indicated redness (*a**) of Golden Delicious and Elstar slices respectively at control conditions and 0 min (immediately applying the treatments). When air exposure was extended beyond 30 min, both cultivars did not show a significant difference of a* (Table 3) which is also shown by Figure 4c,d. Considerably, lower *L** values and higher a* values were observed up to 60 min of air exposure for Golden Delicious slices. Based on the results from ANOVA (Appendix A), the application of AA and CA, alone or in combination, did not significantly affect the *L** value at different air exposure. However, with prolongation of exposure time (i.e., up to 60 min) the *L** value decreased for slices treated with hot-water blanching at 70 °C (Figure 3). The tendency of browning in different cultivars is well perceived because of variation in characteristics of flesh, skin colour and browning potential [62]. Crichton et al., [63] and Delgado-Pelayo et al., [64] mentioned that apples consist of several pigments such as flavonoids that might also be the cause of colour deterioration on the product’s surface. Redness (*a**) of slices from both cultivars did not show significant differences when exposed to air for 30 and 60 min. Thus, in our case, when slices were exposed for extended periods before processing, changes in a* value were not significant regardless of cultivars (Appendix A). The 70 °C HWB treatment was observed to be the worst as it significantly decreased *L** and increased *a** in both cultivars. Eventually, the temperature at 70 °C was sufficient to inactivate the enzymes, while apple slices lost the tissue integrity with cell breakage, facilitating more contact to air and the release of endogenous enzymes. The enzymes ultimately encountered substrates in different cell compartments, resulting to surface colour changes. Moreover, the opening of the cellular pores during the heating process might have led to the induction of secondary product synthesis, including a variety of phenolic compounds. Therefore, blanching treatments can have an adverse effect on cell membranes that might deteriorate in different ways such as changes in composition, structure, function or the loss of protein functionality [65]. Decreases in luminance (*L**) and yellowness (*b**) and increases in redness (*a**) of Golden Delicious apple slices, treated with hot-water blanching at 100 °C for 4 min, were also reported by Lavelli and Caronni, [22]. 

### 3.5. Assessment of Oxidative Enzymes Activity at Optimum Goals

Our findings showed the two apple cultivars to have different PPO/POD enzymes activity. Both specific enzyme activity and substrate availability, such as the total phenolic content, might influence the extent of the apples’ browning [66,67]. 

A low PPO and POD activity was achieved on Golden Delicious slices (Figure 5a) when treated with 70 °C HWB + 1 % AA + 1 % CA and Elstar slices (Figure 5b) with 60 °C HWB + 1 % AA + 1 % CA as shown by the desirability index with a better prediction value.

As expected, 70 °C HWB + 1 % AA + 1 % CA was found to be the optimum treatment to inhibit both PPO and POD activity in Golden Delicious slices (Figure 6 and Figure 7). This is most probably because diffusion processes were enhanced at higher temperatures. According to Rico et al., [68] AA acts more effectively when combined with other pre-treatments such as thermal treatment or citric acid. In this study, AA treatment showed a significant effect on several treated apple slices. The efficacy of AA seemed to control enzymatic browning. It is well known that AA reacts with oxygen and reduces o-quinones produced by PPO-catalysed oxidation of polyphenols, back to dihydroxy polyphenol. These findings are consistent with the results of Chow et al., [69]; Jang and Moon [70]. However, the effect of ascorbic acid is temporary. Lamikanra and Watson [71] found that AA was incapable of inactivating the PPO activity of cantaloupe melon. The reason might be due to oxidation of AA entirely and formation of browning pigment by the accumulation of *o*-quinone [72]. Furthermore, Yemenicioğlu et al., [73] detected polyphenol oxidase enzymes in the Golden Delicious apple cultivar are entirely deactivated at 68 °C which is close to the 70 °C temperature investigated in this study. Moreover, water used in 70 °C heat treatment might be enough to retard gas exchange resulting in oxygen deficiency and accumulation of carbon dioxide [74].

Both PPO and POD were inactivated with steam blanching at 75 °C and 85 °C. However, the resulting *ΔE* in those samples was undesirable. Apparently, PPO denatured in the temperature range of 50–70 °C which may be due to the tertiary structure changes. Therefore, high-temperature blanching is necessary to control the enzymatic browning caused by both PPO and POD enzymes [75]. Meanwhile, brown colour development in apple slices might be ascribed to both ascorbic acid oxidation and Maillard reaction in the presence of heat treatment [53].

As expected, the effect of heat treatments on the POD activity were lower than on PPO activity, irrespective of treatments and air exposure time. However, assessment of POD is essential as it enhances the browning reactions of PPO. Moreover, to monitor the thermal treatment, POD is frequently used as indicator enzymes due to its higher thermal stability and easiness in being assayed. In fact, the POD inactivation allows the reasonable assumption that other quality-deteriorative enzymes are also inactivated [76,77]. Further, POD may be responsible for enhancement of degradation of phenols when coexisting with PPOs [14].

## 4. Conclusions

This study investigated the effect of physical and chemical treatment and combination of these treatment to reduce browning on apple slices at three different air exposure durations at room temperature. Both ‘Elstar’ and ‘Golden Delicious’ showed a similar pattern of minimal colour change at the beginning of the experiment. However, the two apple cultivars (Golden Delicious and Elstar) showed great variation of colour discolouration and sensitivity to the EA with the applied treatments at 30 min and 60 min air exposure at room temperature. The effect of treatments was noticeable significantly only after 30 min of air exposure on both cultivars. From this study, it was clear that each cultivar requires specific treatments to counteract the browning of fresh-cut apple slices. Treatment of apple slices with AA was effective in minimising the discolouration at the beginning of processing but incurred higher browning sensitivity with prolongation of air exposure at room temperature. Higher blanching temperature reduced the EA but did not prevent the surface colour change. At 60 min exposure time, the *L** value decreased for slices treated with hot-water blanching at 70 °C. Thus, the worst results for total colour change and browning index were obtained using the 70 °C HWB in a combination with 1 % AA +1 % CA. However, the effect of HWB was substantial to reduce EA. Accordingly, the anti-browning effect and reduction of EA behaved contrary to each other under the applied treatments. Further investigation should be carried out to assess the effect of these applied treatments and air exposure on final processed products such as dried apples. Moreover, further studies to investigate the effect of different packaging and microbial activity while exposing treated slices at ambient temperature might be of interest. 

## Figures and Tables

**Figure 1 foods-09-00076-f001:**
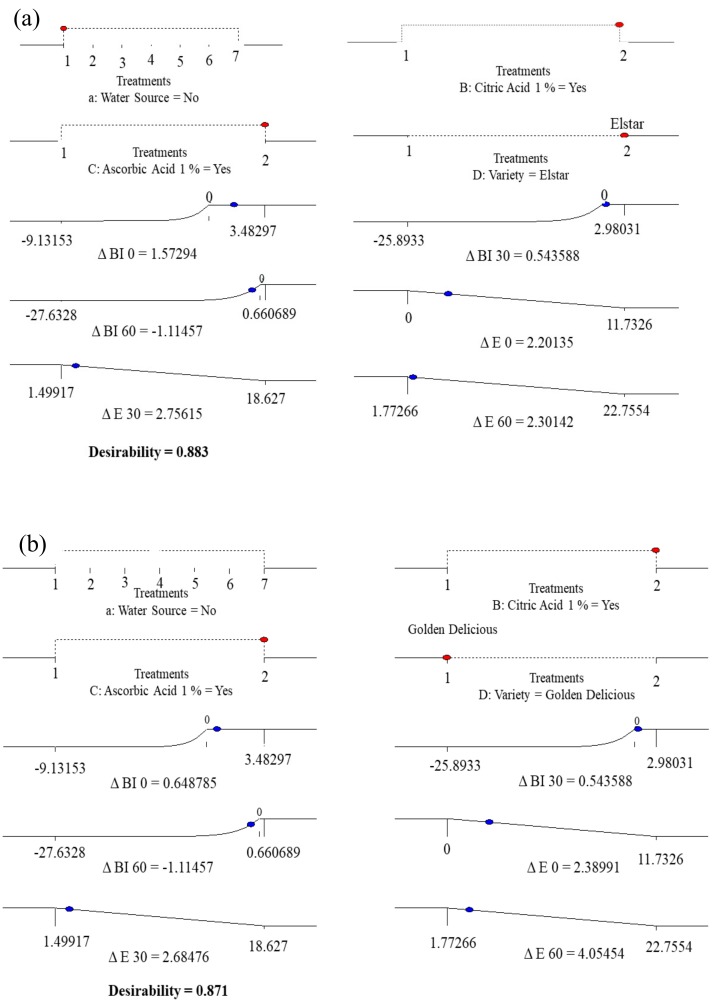
Numerical optimization for surface colour development of Elstar (**a**) and Golden Delicious (**b**) slices subjected to treatments and exposure to air at room temperature for 0, 30 and 60 min. Red dot = factors; blue dot = responses. Climbing up the response ramps means more desirable value.

**Figure 2 foods-09-00076-f002:**
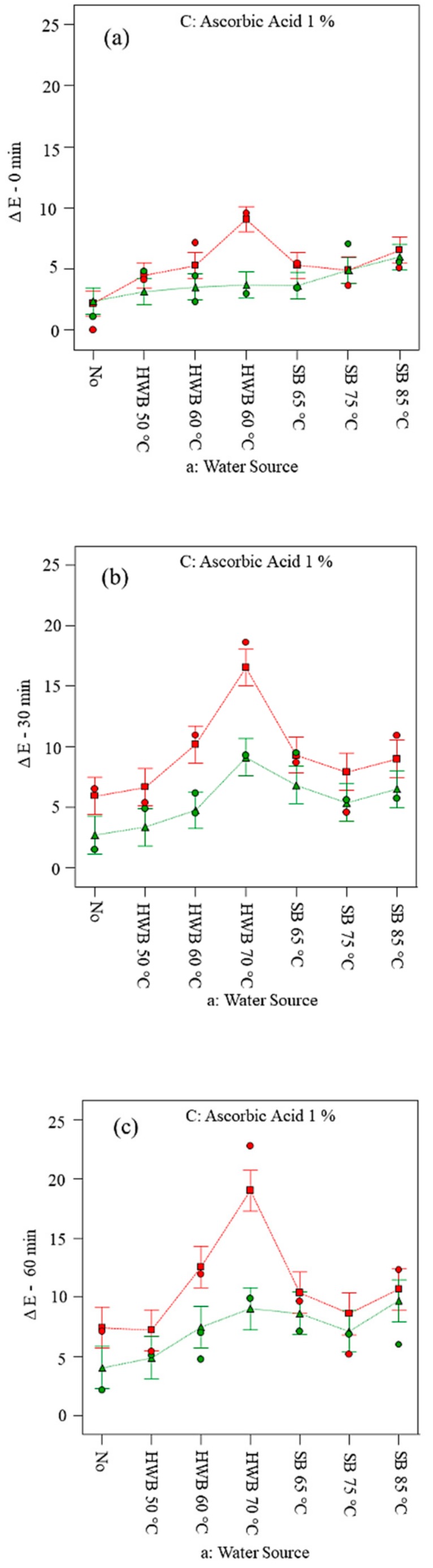
Changes in the CIE1976 colour difference (ΔE) on slices (cv. Golden Delicious) at 0 (**a**), 30 (**b**) and 60 min (**c**) of air exposure. No = no physical and/or chemical treatment (control); HWB = hot-water blanching; SB = steam blanching; X_1_ = a: Water Source; X_2_ = D: Variety; B: Citric acid 1% = No; C: Ascorbic acid 1% = no; red point = D1 Golden Delicious; green point = D2 Elstar.

**Figure 3 foods-09-00076-f003:**
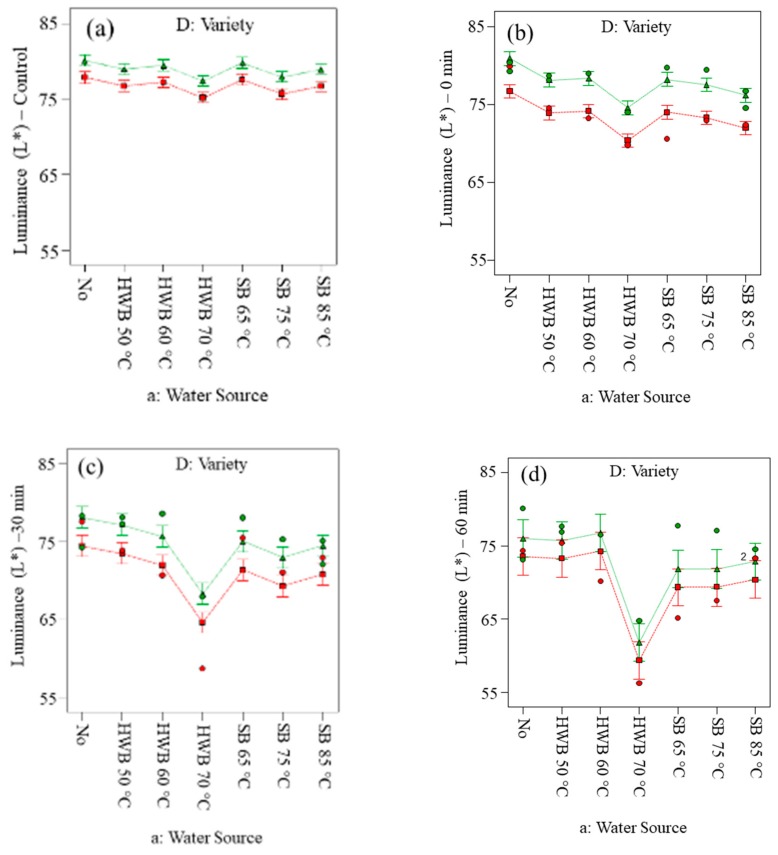
Changes in luminance (*L**) on slices *(cv.* Golden Delicious *[red colour] and Elstar [green colour])* before applying treatments (**a**) and at 0 (**b**), 30 (**c**) and 60 min (**d**) of air exposure. No = no physical and/or chemical treatments (control); HWB = hot-water blanching; SB = steam blanching; X_1_ = a: Water Source; X_2_ = D: Variety; B: Citric acid 1% = No; C: Ascorbic acid 1%= Yes; red point = D1 Golden Delicious; green point = D2 Elstar.

**Figure 4 foods-09-00076-f004:**
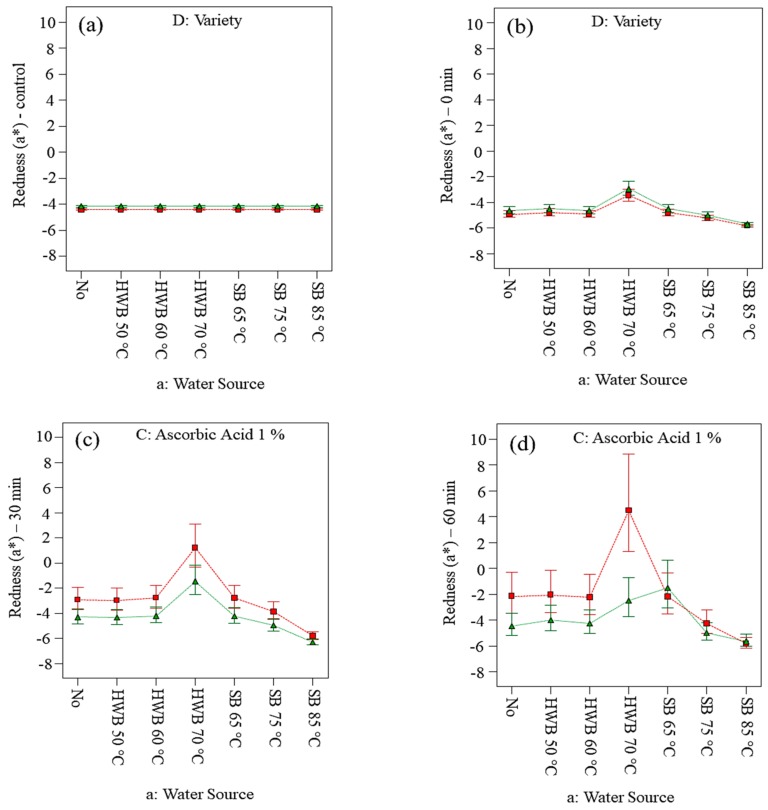
Changes in the redness (*a**) on slices (cv. Golden Delicious and Elstar) before applying treatments (**a**) and at 0 (**b**), 30 (**c**) and 60 min (**d**) of air exposure. No = no physical and/or chemical treatments (control); HWB = hot-water blanching; SB = steam blanching; X_1_ = a: Water Source; X_2_ = D: Variety; B: Citric acid 1% = No; C: Ascorbic acid 1%= Yes; red point = D1 Golden Delicious; green point = D2 Elstar.

**Figure 5 foods-09-00076-f005:**
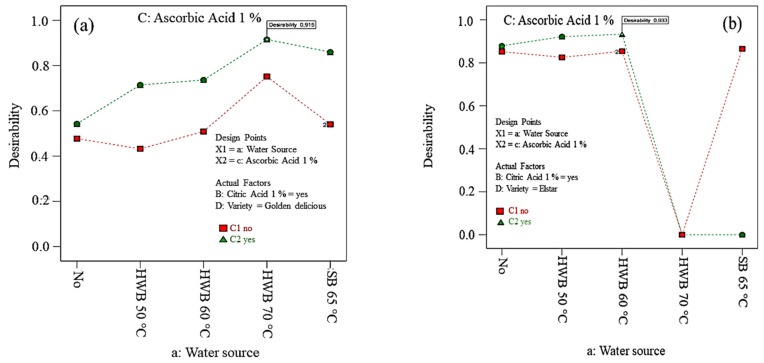
Optimum treatment for both polyphenol oxidase (PPO) and peroxidase (POD) enzymes activities on Golden Delicious (**a**) and Elstar (**b**) slices expressed as desirability index. No = no physical and/or chemical treatments (control); HWB = hot-water blanching; SB = steam blanching.

**Figure 6 foods-09-00076-f006:**
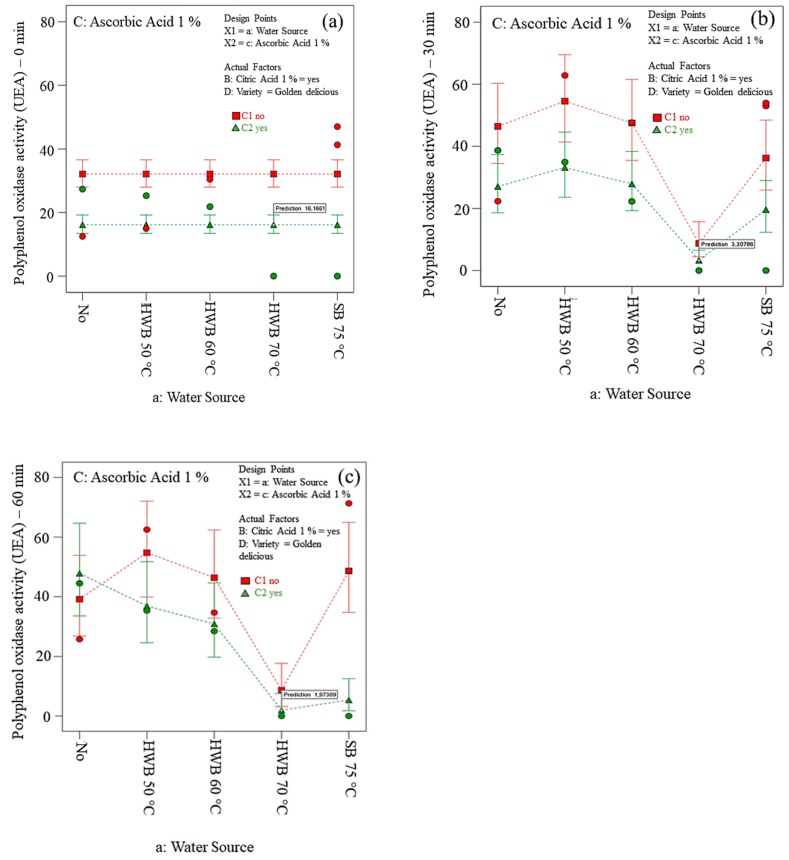
Polyphenol oxidase activity on treated slices *(cv.* Golden Delicious*)* exposed to air at room temperature for 0 (**a**), 30 (**b**), and 60 min (**c**). No = no physical and/or chemical treatments (control); HWB = hot-water blanching; SB = steam blanching.

**Figure 7 foods-09-00076-f007:**
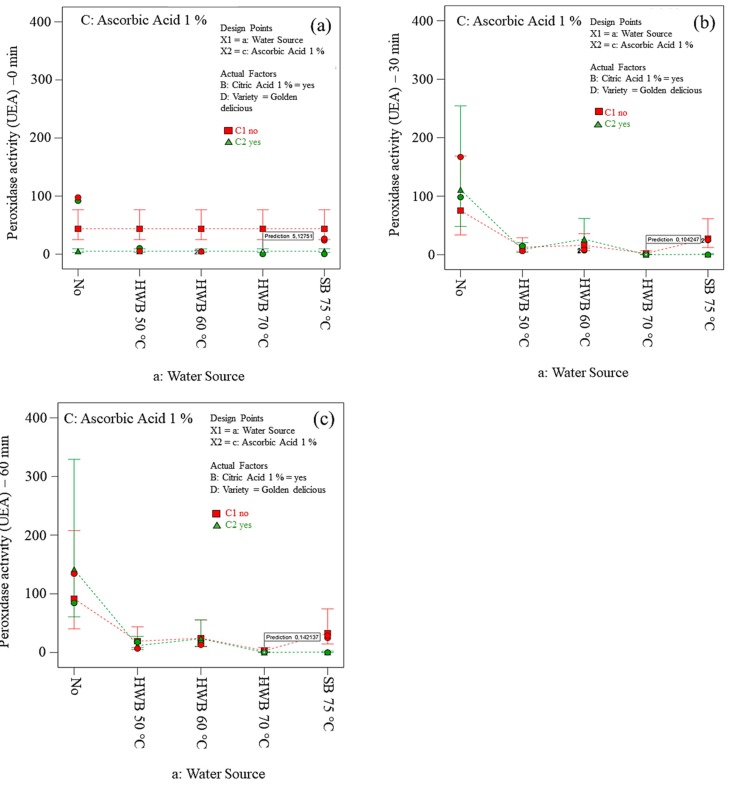
Peroxidase activity on treated slices (*cv.* Golden Delicious) exposed to air for 0 (**a**), 30 (**b**), and 60 min (**c**). No = no physical and/or chemical treatments (control); HWB = hot-water blanching; SB = steam blanching.

**Table 1 foods-09-00076-t001:** Description of different treatments applied on different cultivars based upon previous research.

Apple Cultivar	Pre-Treatments	Conditions	Parameters Measured	Results	Source
Fuji	(i) 0.5 % CA (ii) exposed to UV-C lamp; (iii) dipped in CA + exposed to UV-C; (iv) Control—5 min dipping time	-After treatment, fresh-cut apples were placed in plastic foam tray (15 × 21 × 2.5 cm) and wrapped with PE cling film and stored at 5 ± 2 C for 15 days	- TPC - Phenolic compound - Δ E - PPO	- CA has negative impact on Δ E compared to control - UV and UV + CA: reduce BI - UV, CA and CA + UV treatments: inhibit PPO activity (*p* < 0.05) - UV + CA: control BI, lowering the microbial activity and PPO activity	[18]
Braeburn	(i) 55 °C Hot water (HW) treatment (ii) 65 °C HW- treatment for 30 s—immersed in AA/CA solution (40 g AA and 20 g citric acid in 1 L deionized water) for 5 min	Apple slices were stored immersed in sugar syrup in pails for up to 13 days	-Tissue strength - Browning index (BI) -Total soluble solid (TSS) -Titratable acidity (TA) -Vitamin C -Microbial analysis	-HWT at both 55 °C and 65 °C had positive effect on important quality parameters such as TSS, TA or vitamin C	[19]
Fuji apple —1.5 cm cubes	(i) 0.5 % Ascorbic acid (AA); (ii) 0.5 % cysteine (CS); (iii) Distilled water (DW); iv) 0.01 % Chlorinated water (CW)—2 min dipping time	- Stored in low density polyethylene (LDPE) bags without sealing in the dark at 4 °C, 90 % RH for 7 days	-Polyphenol oxidase activity (PPO) - Total phenolic content (TPC) - Browning index (BI) - Color changes (Δ E)	− 0.5 % CS: to control PPO activity and BI - CS and AA: lower the TPC	[20]
Golden delicious (GD); Cripps Pink (CP) —1 cm	(i) control; ii) 1% *w*/*v* AA and 0.2 % *w*/*v* Citric acid (CA) with/without ultrasound; iii)1 % *w*/*v* Ca-ascorbate with/without ultrasound—3 min dipping time	- After the treatments, pieces were immediately packaged in 20 × 20 cm bags with modified atmosphere	- Δ E	- Combination of AA and CA with ultrasound: to reduce Δ E - Ca- ascorbate: best treatments - GD shows endure browning better than CP	[21]
Golden delicious	(i) Blanched in a deionized water bath at 100 °C—4 min; (ii) Control /unblanched	- Freeze dried and stored in a dark under vacuum at − 20 °C until used	- Δ E - antioxidant content - antioxidant activity	-Control sample: better Δ E > blanched sample stored at 40 °Cat a_w_ < 0.32 -High PPO in unblanched apples stored at the 0.56 a_w_ level at 20 °C - Catechin is the most unstable phenolic ompounds in both conditions	[22]
Golden delicious —14 mm	(i) AA; (ii) CA; (iii) sodium chloride (NaCl) (iv) potassium metabisulfite (PBS); (v) 1 g/L AA + 1 g/L CA; (vi)PBS + CA; (vii) AA + NaCl	-Samples were dripped for 2–3 min and kept at 2 °C prior to analysis	- PPO activity	- 1 % AA + 0.2 % CA: best treatment - 0.05 % NaCl + 1% AA: completely inhibits the PPO enzymes	[23]
	(a) AA 1 g/L and CA 1 g/L; AA 1 g/L and CA 2 g/L; AA 10 g/L and CA lg/L; AA 10 g/L and CA 2 g/L; (b) PBS 0.1 g/L and CA 1 g/L; PBS 0.1 g/L and CA 2 g/L; PBS 0.3 g/L and CA 1 g/L; PBS 0.3 g/L and CA 2 g/L; (c) AA 10 g/L and NaCl 0.2 g/L; AA 10 g/L and NaCl 0.5 g/L; AA 10 g/L and NaCl 1 g/L (d)control dipped in deionized water—5 min			- Individual application of concentrations between 0.2-10 g/L of AA, CA, NaCl has a low impact on inhibiting the PPO activity -10 g/L AA + 2 g/L CA: inhibited 87 % the PPO activity - 10 g/L AA + 0.5 g/L NaCl: completely inhibits the PPO activity	
Granny Smith	(i) 10 µL/L nitric oxide (NO) exposure; (ii) Diethylenetriamine/nitric oxide, (DETANO) treatment —2 hour	-Stored at 5 °C	-PPO activity	-Effect of treatments is DETANO < NO gas < water < untreated - PPO activity increased with storage period (6 months < 3 months < 0 months)	[24]
Granny Smith	(i) CA + Ca- ascorbate + NAC, and CA +AA +NAC. Tested levels for AA/Ca ascorbate were 0–7% with 3.5% as center points; and for NAC, the levels tested were 1–2%—3 min dipping time	-Placed into zipper sealed bags (15.2 × 12.7 cm) and stored at 4 C for 21 days	-Colour CIELAB parameters	-Combinations of 4% CA, 3–4 % AA and 1.5–2.0% NAC: the best ombination pre-treatments	[25]
Idared, Golden Delicious, Gala, Gloster, Cripps Pink, Braeburn, and Fuji—1 cm thick	(a) Control; (b) AA (1%, *w*/*v*) and CA (0.2%, *w*/*v*); (c) AA (1%, *w*/*v*)1NaCl (0.05%, *w*/*v*); (d) NaCl (1%, *w*/*v*); (e) CA (1%, *w*/*v*), and (f) Ca-ascorbate (1%, *w*/*v*)	-Apple slices storage times (0, 30, 180, 1440 min)	- Δ E -BI	-Fresh-cut processing act differently affected different apple cultivars -GD and Cripps Pink showed the least Δ E and BI - Air exposure showed largest influence on BI regardless of cultivar from 60 to 180 min storage -CIELab variables remained unchanged during 60–180 min of storage. - Ca-ascorbate: best antibrowning solution	[26]
Royal Gala—1 cm	(i) 5 g/L Carrageenan, 20 g/L alginate, Exopolysaccharide (EPS) 5 g.L^–1^, EPS 10 g.L^–1^, 5 g/L pectin or 2 g/LCMC solutions—2 minutes dipping time	-Stored in the boxes (polypropylene, l * w * h: 105 * 75 * 55 mm). All the boxes were covered with a lid, but not closed to not create a modified atmosphere; then, they were stored at 4°C	-Colour CIELAB parameters -PPO activity	-A good correlation between the colour and the PPO activity of the coated apple cubes. -Pectin solution was the best coating to reduce the PPO activity	[27]
Jonagored red—1.5 cm	(i) 42.6 mM AA (0.75% *w*/*v*); (ii) 21.3 mM AA + 33.8 mM CC and 14.2 mM AA + 22.5 mM CC + 13.0 mM CA	-Cubes were stored in open glass jars at 4 °C and atmospheric pressure for 7 days in the dark	-Colour CIELAB parameters - BI - TPC - PPO	-AA: best treatment - Large correlation were found between colour parameters and the total phenolic compounds -A negative correlation observed between L* value and BI -No correlations were observed between PPO activity, colour parameters, BI or TPC	[28]
Red delicious (RD) and Granny Smith (GS)	(i)100 mg/L ClO_2_; (ii)100 mg/L ClO2 + 3 % AA	-Each sampling time at 0, 48, 96 h after cutting were stored at 4 °C	-PPO -POD -BI	-At 0 h, BI in GS is lower than RD -At 48 h, ClO^2+^ AA had lower BI, inhibit PPO and POD activity on both RD and GS slices -GS has lower PPO activity compared to RD	[29]

**Table 2 foods-09-00076-t002:** Factors and responses included in the experimental design.

Factor name	Factor type	Levels	Description
Water Source	Hard-to-change	Control	No treatment
		HWB ^a^ 50 °C	Hot-water blanching at 50 °C
		HWB ^a^ 60 °C	Hot-water blanching at 60 °C
		HWB ^a^ 70 °C	Hot-water blanching at 70 °C
		SB ^b^ 65 °C	Steam blanching at 65 °C
		SB ^b^ 75 °C	Steam blanching at 75 °C
		SB ^b^ 85 °C	Steam blanching at 85 °C
Apple variety	Easy-to-change	Elstar	
		Golden Delicious	
Dipping in CA ^c^	Easy-to-change	Control	0% (*w*/*v*) citric acid
		1% CA ^c^	1% (*w*/*v*) citric acid
Dipping in AA ^d^	Easy-to-change	Control	0% (*w*/*v*) ascorbic acid
		1% AA ^d^	1% (*w*/*v*) ascorbic acid

HWB ^a^—hot-water blanching; SB ^b^—steam blanching; CA ^c^—citric acid; AA ^d^—ascorbic acid.

**Table 3 foods-09-00076-t003:** Chemical characteristics of the apple fruits of the two varieties studied in the experimentation.

Parameters	Cultivar	
	Golden Delicious	Elstar
TSS (%)	11.6 ± 0.2 ^a^	13.16 ± 0.15 ^b^
pH	3.74 ± 0.02 ^a^	3.39 ± 0.03 ^b^
TA (meq malic acid/100g FW)	2.23 ± 0.24 ^a^	4.11 ± 0.15 ^b^
TSS/TA	5.23 ± 0.52 ^a^	3.21 ± 0.14 ^b^

Results are presented as mean ± standard deviation of the mean. Row mean values followed by the same letter of significance are not different according to LSD (*p* < 0.05).

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
