# Peer review of "Optimisation of Physical and Chemical Treatments to Control Browning Development and Enzymatic Activity on Fresh-cut Apple Slices"

_foods, 2020, doi:10.3390/foods9010076_

Round 1

Reviewer 1 Report

All figures must be rewritten, which figure resolution, words, and units could not clearly to read and to understand. Generally, the majority (about 70%) of references must be the most recent (of the last 5 years), but this paper is only up to 33 %

Author Response

1. Does the introduction provide sufficient background and include all relevant references?- Can be improved

R: The introduction has been improved and updated with recent studies.

2. All figures must be rewritten, which figure resolution, words, and units could not clearly to read and to understand. Generally, the majority (about 70%) of references must be the most recent (of the last 5 years), but this paper is only up to 33 %

R: All figures have been updated with 600 dpi resolution, the font size has been increased. Few of the recent studies on the application of pre-treatment on apple slices are added in the text (Kim 2018; Yildiz 2019; Jukanti 2017; Li 2019; Cofelice 2019; Han 2019; Rux 2019; Iqbal 2019; Tylewicz 2019).

However, it is difficult to find the recent studies relevantly done on PPO activity in apple slices. This study presents and compares the results from previous research that were carried out only on apple.

Reviewer 2 Report

This research including topic, methods and conclusions is not innovative except for statistical analysis and the difference specification.  But the narrative writing was in a state of disorder.  My suggestions to deal with the problem are as follows:

Carefully use scientific concept to reconstruct the whole article. Delete or simplify unnecessary part or word to make the text brief and simple. Many papers were reviewed in Table 1. However, the shortcomings and merits were not concluded and linked to the research design. The statistical analysis and result expression seemed to be the unique feature of the study. The text content and figures should be clearly and definitely descripted.  In addition, technical terms, acronym, small and capital letters, and even unity of references should be correct.

Author Response

1. Extensive editing of English language and style required 

R: The English has been edited by a native speaker

2. Does the introduction provide sufficient background and include all relevant references?

R: The whole introduction section has been revised and more information has been added with the recent studies.

3. Is the research design appropriate?

R: Split-plot design was chosen to conduct the experimentation because it is one of the oldest approaches use in agricultural experiments and originally developed by Fisher (1925). A split-plot experiment is a blocked experiment, where the blocks serve as experimental units for a subset of the factors. There are two levels of experimental units. The blocks are referred to as whole plots, while the experimental units within blocks are known as split plots or split units. There are two levels of randomization within the two levels of experimental units. A true replication in the whole plot factor is present in conducting a split-plot experiment.

The advantage of the spilt- plot design is factors are often differentiated with respect to the ease with which they can be changed from experimental run to experimental run. This design is often superior to completely randomized designs in terms of cost, efficiency, and validity.

[Jones, B., & Nachtsheim, C. J. (2009). Split-plot designs: What, why, and how. Journal of quality technology41(4), 340-361.]

4. Are the methods adequately described?

R: The description of the methods has been revised. Section 2.3, 2.4, 2.8 and 2.9.

5. Are the conclusions supported by the results?

R: The conclusions section has been reformulated to support the results.

6. Carefully use scientific concept to reconstruct the whole article. Delete or simplify unnecessary part or word to make the text brief and simple. Many papers were reviewed in Table 1. However, the shortcomings and merits were not concluded and linked to the research design. The statistical analysis and result expression seemed to be the unique feature of the study. The text content and figures should be clearly and definitely descripted.  In addition, technical terms, acronym, small and capital letters, and even unity of references should be correct.

R: The whole article has been reconstructed. The uniformity of technical terms, acronym, small and capital letters and even unity of references has been considered and corrected. The shortcomings and merits from the previous studies were linked to the present research, lines 73-83.

Now, the figures resolution, fonts, and the index has been improved.

Reviewer 3 Report

Generally, presented manuscript is well written and contains some useful information concerning the  pre-treatments methods to reduce the browning development of apple slices from two cultivar. The novelty of presented paper is rather average.

Moreover, I have some comments and remarks:

It is inadequate to cite references in the abstract.

Line 59 (EA) ??? is not for sense

Line 64. Why “Enzymatic” is form capital letter?

The second formula x for chromatically coordinate should be numbered?

Authors included a comprehensive description of statistical evaluation of data. However in the most cases  it is no information how many repetitions of individual test were performed.

Some part of text is repeated see lines 227-240 and 245-258.

Where is supplementary material (table S.4)?

Why authors connected points with lines on the charts? Is not adequate in this case.

The font size of charts is too small.

Author Response

1. English language and style are fine/minor spell check required

R: The whole manuscript has been revised in terms of language editing.

2. Are the methods adequately described?

R: The description of the method has been revised. Section 2.3, 2.4, 2.8 and 2.9.

3. Are the results clearly presented?

R: All the results has been revised.

4. It is inadequate to cite references in the abstract.

R: The same bracket was used to denote the serial number. The number has been removed from the abstract and () is used instead of [] to make distinguishable from the references. Line 116-117.

5. Line 59 (EA) ??? is not for sense

R: EA has been removed, line 42.

6. Line 64. Why “Enzymatic” is form capital letter?

R: It has been revised to a small letter.

7. The second formula x for chromatically coordinate should be numbered?

 R: The number has been given as eq 2, line 140.

8. Authors included a comprehensive description of statistical evaluation of data. However, in the most cases it is no information how many repetitions of individual test were performed.

R: Revised and the number of samples is added, line 102-105.

9. Some part of text is repeated see lines 227-240 and 245-258.

R: The repeated text has been removed.

10. Where is supplementary material (table S.4)?

R: It was not possible to attach the supplementary material during the submission process. However, it is added at the end of the revised manuscript.

11. Why authors connected points with lines on the charts? Is not adequate in this case.

R: The authors already tried to plot point in way of no connecting lines, however, we realised that using connected points give clearer expression to readers.  

12. The font size of charts is too small.

R: All the figure font size has been increased.

Round 2

Reviewer 3 Report

Manuscript was significantly improved and it is adequate for publication.